# Video Laryngoscopy Using King Vision™ aBlade™ and Direct Laryngoscopy in Paediatric Airway Management: A Randomized Controlled Study about Device Learning by Anaesthesia Residents

**DOI:** 10.3390/jcm11195676

**Published:** 2022-09-26

**Authors:** Katharina Epp, Sophie Zimmermann, Eva Wittenmeier, Marc Kriege, Frank Dette, Irene Schmidtmann, Nina Pirlich

**Affiliations:** 1Department of Anaesthesiology, University Medical Centre of the Johannes Gutenberg-University, Langenbeckstr. 1, 55131 Mainz, Germany; 2Institute of Medical Biostatistics, Epidemiology and Informatics, University Medical Centre of the Johannes Gutenberg-University, Langenbeckstr. 1, 55131 Mainz, Germany

**Keywords:** airway management, paediatric, video laryngoscopy, endotracheal intubation

## Abstract

Background: Airway management in children is challenging due to anatomical and physiological differences. This randomized trial investigates whether anaesthesia residents can intubate the paediatric trachea more quickly and with a higher success rate using the King Vision™ Paediatric aBlade™ video laryngoscope (KVL) compared to conventional direct laryngoscopy (DL). Methods: Eleven anaesthesia residents (mean age: 31 years, mean training status 47 months) were each asked to perform intubations with the KVL and DL in paediatric patients. The primary outcome was the first-attempt success rate. Secondary outcomes were the time to best view (TTBV), time to placement of the tracheal tube (TTP), time to ventilation (TTV), and participant-reported ease of use on a Likert scale. Results: 105 intubations with the KVL and 106 DL were performed by the residents. The success rate on the first attempt with the KVL was 81%, and the success rate on the first attempt within a given time limit of 30 s was 45%, which was lower than with DL (93% and 77% with time limit, *p* < 0.01). The median TTBV [IQR] on the first attempt with KVL was 7 [5–10] s, the median TTP was 28 [19–44] s, and the median TTV was 51 [39–66] s. DL-mediated intubation was significantly faster (TTP: 17 [13–23] s; *p* < 0.0001 and TTV: 34 [28–44] s; *p* < 0.001). Application of the KVL was rated as difficult or very difficult by 60% of the residents (DL: 5%). Conclusion: In contrast to promising data on the paediatric training manikin, residents took longer to intubate the airway in children with the KVL and were less successful compared to the DL. Therefore, the KVL should not be recommended for learning paediatric intubation by residents.

## 1. Introduction

Intubation of the paediatric airway is challenging due to specific anatomical differences compared to the adult patients. These include the position of the larynx, the omega-shape of the epiglottis, and the different, cephalad angle of the vocal cords [1,2]. Additionally, children have a lower tolerance to apnoea, so intubation should be performed very quickly. Successful intubation of paediatric patients therefore requires both experience and appropriate tools. The ideal equipment to successfully train tracheal intubation in children is still unknown. Most paediatric laryngoscopes are scaled-down versions of adult video laryngoscopes and only few devices are specifically adapted to the anatomy of the paediatric larynx.

Studies suggest that using video laryngoscopy to secure the airway in both adults and children is beneficial to the patient [3,4,5,6,7,8,9,10,11,12,13,14,15,16,17,18]. The King Vision™ Paediatric aBlade™ video laryngoscope (KVL) was introduced in 2017 and was specifically designed for the paediatric airway and intubation training. The equivalence or superiority of KVL when used on manikins and adult patients compared to direct laryngoscopy or other techniques has been demonstrated [19,20,21,22,23,24,25,26]. However, the applicability and specific learning curve of the KVL for paediatric intubation has only been investigated in two studies evaluating the KVL in children under 2 years of age [6,27]. However, the intubations were only conducted by experienced anaesthesiologists who had performed more than 1000 intubations with the Miller blade in paediatric patients but had no experience with the KVL. Therefore, it remained unclear whether the KVL is superior to the Miller blade for paediatric airway management even among anaesthesia residents. A previous study investigated the intuitive use of the KVL on a paediatric training manikin. We showed that participants needed only five successful tracheal intubations with the KVL to achieve a 100% success rate for the first tracheal intubation attempt [28].

In the present prospective, randomized, single-centre study, we aimed to investigate how quickly and how reliably anaesthesia residents can learn to intubate the trachea of paediatric patients using KVL. Based on our preliminary results obtained with manikins, we hypothesized that intubation can be learned quickly by anaesthesia residents using the KVL.

## 2. Materials and Methods

Following approval by the local research Ethics Committee (Ethics Committee of the Medical Association of the Rhineland-Palatinate state, Germany, approval number: 2018-13185), children requiring elective surgery with the indication for tracheal intubation were included in this prospective, controlled, randomized, and registered study (ClinicalTrials.gov, NCT03571295). Written informed consent was obtained from the legal guardians of all participating children. Eligibility criteria included an age between 1 and 10 years and an American Society of Anesthesiologists (ASA) classification of 1 or 2. Exclusion criteria were a known difficult airway and increased risk for pulmonary aspiration (e.g., insufficient fasting for solids < 6 h, pathophysiological delay in stomach emptying, and intestinal obstruction). The recruitment took place between July 2018 and June 2020.

Eleven anaesthesiologists in training who had been in residency for a minimum of 24 months and a maximum of 60 months were selected for this study and their written informed consent was obtained. None of the trainees had previously used the KVL in a patient.

The King Vision™ Paediatric aBlade™ (Ambu^®^ GmbH, Bad Nauheim, Germany) video laryngoscope with unchanneled blade size 2 was used, according to the manufacturer’s recommendation for children from 1 to 10 years of age (Figure 1).

In preparation for the study, the use of the KVL was demonstrated by presenting a 2 min video with instructions. Each trainee performed at least 7 intubations with the KVL and 6 direct laryngoscopies with the Macintosh blade (DL) but no more than a total of 20 intubations with both devices to assess his or her ability to successfully intubate a paediatric patient. The device for the first tracheal intubation of each trainee was randomly assigned to either the KVL or DL group using a web-based service (QuickCalcs, GraphPad Software, La Jolla, CA, USA). Devices were then used in alternating order. Upon admission to the operating room, standard monitoring methods including ECG, non-invasive blood pressure, and pulse oximetry were applied. The children’s heads were kept in a neutral position via a gel head ring. Preparation of intubation and the use of neuromuscular blocking drugs to improve laryngeal view were performed according to standard operating procedures of the University Medical Centre of the Johannes Gutenberg University Mainz, Germany. All trainees performed both intubation with the KVL (n = 105) and DL (n = 106) interchangeably in consecutive patients. For intubation with the KVL, a malleable stylet, angulated into a hockey-stick shape, was inserted into the endotracheal tube. Choice of anaesthetic was not affected by the study. Anaesthesia was induced, and once the reflex of the eyelid had expired and an adequate level of anaesthesia was achieved, neuromuscular blocking drugs (Atracurium and Mivacurium) were administered; ventilation was performed via a face mask. Intubation was performed after onset of the neuromuscular blocking. Airway manoeuvres (optimal external laryngeal manipulation manoeuvre, OELM, and positioning of the head) could be used as needed to improve laryngeal view. Different tube types were used, according to the surgical requirements.

The primary outcome was the success rate of intubation in the first attempt. A time limit was set depending on a change in device or confirmation by the attending supervisory anaesthesiologist to continue the attempt with a given device. The first attempt of tracheal intubation was considered successful if the total time of intubation did not exceed 30 s for the KVL and 25 s for DL [29]. The first attempt of intubation was considered a failure if the aforementioned times were exceeded or if oesophageal intubation or oxygen desaturation <90% occurred. The attending supervisor decided whether the chances of success for the current tracheal intubation attempt after expiration of the time limit were adequate to continue or if termination and a switch in user and/or device was necessary. Due to this fact, we evaluated a first attempt success rate with and without a time limit. Outright failure was defined as inability of the trainee to establish a sufficient airway within two additional intubation attempts, using either a laryngoscopic device of his or her choice or a supraglottic airway device.

Secondary outcome was the time needed for the intubation steps in the first attempt. Time was measured by an independent observer using a stopwatch. A total of three times during the intubation process were analysed: the time to best view (TTBV) was defined as the time interval between the blades entering the mouth to confirmation of the best glottic view. The time to placement of the tracheal tube (TTP) was determined as the time until the black mark on the tracheal tube passed the vocal cords. Subsequently, the time to ventilation (TTV) was measured via confirmation of end tidal capnography. Trainees rated the usability of the intubation device on a Likert scale from very easy to very difficult. We differentiated between the occurrence of complications and difficulties during the intubation process. Complications included oesophageal intubation, desaturation of more than 2% from the initial value, and bleeding of airway mucosa. Difficulties were wrong tube size, problems placing the tube, guide wire problems, problems passing the tube past the blade (blade size), and problems with visual quality (e.g., fogging, contrast, darkness).

When determining sample size, we aimed to have at least 80% power to establish an overall 15% difference in success rates at the 5% significance level assuming that the lower success rate would be 80%. This would be possible with 76 independent observations per type of laryngoscope when using a chi-squared test. As multiple observations per provider would lead to dependent observations, thus losing some power, we decided to increase sample size to 100 observations per type of laryngoscope and a maximum of 10 observations per provider within each group.

For the primary endpoint, the OR was calculated with 95% CI. A generalized estimating equations logistic regression was performed to evaluate the progress of each trainee with increasing number of intubations. The results from the model are displayed in a learning curve, combined with the observed proportion of successes at each intubation number.

For the secondary endpoint, intubation times were compared using a Wilcoxon test stratified by number of intubation. Further categorical endpoints were described by occurrences and descriptive *p*-values. A GEE (generalized estimating equations) model for binary and multinomial endpoints with logit link was used to accommodate the fact that users were observed multiple times. Both type of laryngoscope and number of intubations were included as covariates in the model, so that both effects of type of laryngoscope and time trends on success rate could be assessed. Likert scale was also analysed via GEE model as normally distributed variable with identity link, device, and intubation number as covariables.

Statistical analyses were performed using SPSS version 23 (IBM Corp., Armonk, NY, USA) and SAS 6.4 (SAS Institute Inc., Cary, NC, USA).

## 3. Results

### 3.1. Patient and Trainee Characateristics

A total of 371 children were assessed for eligibility to participate in the study, of which 211 children received the allocated intervention and were finally analysed. 105 children were intubated using the KVL, 106 children with DL (Figure 2).

Patient and trainee characteristics were comparable (Table 1). There were no outright failures.

### 3.2. Primary Endpoint–First Attempt Success Rate with and without Time Limit

For the primary endpoint of the present study, the first attempt success rate without time limit for the KVL group was 81% (n = 85), and within the given time limit it was only 45% (n = 47). To demonstrate that the trainees are in principle capable of intubating a paediatric patient, we evaluated the first attempt success rate of DL. The analysis of the success rate without the given time limit for DL was 93% (n = 99) and within the given time limit 77% (n = 82). The GEE model yielded significant differences in success rates between DL and KVL (*p* < 0.0001, both for success on first attempt with and without time restriction). OR = 0.226 (95% CI = [0.144; 0.356]) for success with time restriction comparing KVL to DL. When considering success on first attempt without time restriction OR = 0.300 (95% CI = [0.102; 0.979]) was observed when comparing KVL to DL. The observed results and the predicted results from the GEE model, which was performed to evaluate the progress of each trainee as the number of intubations increased, are displayed in a learning curve, without and with time limit in Figure 3A,B.

### 3.3. Secondary Endpoint–Time Needed for the Intubation Steps

The secondary endpoint was the time needed for the intubation steps, with sub analyses conducted for the time to best view (TTBV), time to placement of the tube (TTP) and time to ventilation (TTV) measured via confirmation of end tidal capnography. The results are shown in Figure 4.

Whereas median TTBV was 7 s for both devices (DL: 7 [5–9] s, KVL: 7 [5–10] s), the median TTP was 28 [19–44] s for KVL and 17 [13–23] s for DL (*p* < 0.0001) and the median TTV was 51 [39–66] s for KVL and 34 [28–44] s for DL (*p* < 0.001). The median TTP was 11 s longer in the KVL group compared to the DL group (*p* < 0.0001), and the median TTV was 17 s longer in the KVL group compared to the DL group (*p* < 0.0001).

### 3.4. Further Endpoints–Intubation Difficulties and Usability of Device

KVL use was associated with more intubation difficulties than DL (KVL: 34% vs. DL: 8%, *p* < 0.0001). 60% of the anaesthesia residents rated the usability of intubation with KVL as difficult or very difficult to handle. Usability of intubation with the DL was rated as difficult or very difficult to handle by 5% of the trainees. Further intubation characteristics are listed in Table 2.

## 4. Discussion

Learning to intubate the paediatric airway is a major challenge for anaesthesia residents. The KVL is a recent developed video laryngoscope that promises to be perfectly adapted to the paediatric airway and thus allow optimal intubation conditions. The present study investigated how reliably and quickly anaesthesia residents can learn to intubate the trachea of paediatric patients with the KVL compared to the DL. Surprisingly, anaesthesia residents learned intubation significantly worse with the KVL, as evidenced by a lower success rate, longer intubation times, and more intubation complications compared to the DL. These results differ from our expectations, which were based on the promising results of our paediatric manikin study, which had shown that the use of the KVL for airway intubation in children can be easily learned by anaesthesia residents [28]. In the light of the results of the present study, the recommendation for residents to learn airway intubation in children with the KVL may need to be reconsidered.

The present study reveals that the use of the KVL decreases the success rate of the first attempt to intubate the paediatric airway within a given time limit. Only 45% of the trainees were able to successfully intubate the infant trachea on the first attempt, whereas in our previously conducted paediatric manikin study, one attempt with the KVL was sufficient to apply the technique with a 100% success rate. Furthermore, in the manikin study, all trainees rated the use of the KVL as “very easy,” whereas in the real-life setting of the present study, more than half of the trainees rated the intubation with the KVL as “difficult” or “very difficult”. These different results may be explained by the fact that the trainees in this study did not have the opportunity to practice with the new device and that the intubations with KVL were not performed consecutively but over the time of several weeks. It also became obvious that the handling of a simulated airway, especially in children, cannot be compared to a real-life scenario.

The suitability of the KVL in paediatric clinical practice has not yet been thoroughly evaluated. As far as we know, there exist only two publications evaluating the use of the KVL in children in a clinical setting [6,27]. Jagannathan and colleagues compared 200 paediatric intubations in children under 2 years of age between KVL and DL using a Miller blade [6]. All 200 (100/100) intubations were performed by one of the five experienced study investigators, each of whom had performed more than 1000 tracheal intubations with Miller blades in young children. The success rate of the first attempt was 98% for Miller blade versus 94% for KVL (*p* = 0.28). Colleagues from India revealed a success rate of the first attempt of 100% and similar terms of time for the KVL and DL using a Macintosh blade for elective tracheal intubation in 78 children of age less than 1 year [27]. KVL was associated with superior glottic visualisation, better ease of intubation and lower intubation difficulty score. All intubations were performed by an experienced anaesthesiologist, who had done at least 30 intubations with each device. These results contrast with our own findings. Although the children in Jagannathan et al.’s study were younger and smaller, with a median age of 9 months and a median weight of 9 kg, the success rate in the KVL group was higher than in our own study. Assuming that the intubation of younger and smaller children is more difficult, these differences could be explained by the fact that intubation in the present study was performed by trainees rather than experienced anaesthesiologists. The trainees who participated in our study had intubated less than 100 children, whereas the experts who participated in study by Jagannathan et al. had performed more than 1000 such intubations. While the use of the KVL is straightforward for experienced anaesthesiologist, anaesthesia residents would probably better of using the DL to learn how to secure the paediatric airway. Furthermore, the KVL had never been used before by our trainee anaesthesiologists, neither in children nor in adults, while in the study by Jagannathan et al., some of the providers had used the device before.

Which intubation device is the most appropriate for paediatric patients in terms of resident learning curve remains a matter of discussion. Only one publication noted an improvement in the overall success rate of endotracheal intubation in children with a training, although the first attempt success rate did not improve [30]. However, this study evaluated approximately 22 intubations per user over a 3 years’ period and was conducted in an intensive care setting. Compared to endotracheal intubation in the operating suite, endotracheal intubation in the intensive care setting clearly has more challenging factors such as frequent difficult airway features, unstable haemodynamics, existing respiratory failure, and urgent and emergent nature. These patient factors make endotracheal intubation in the intensive care setting more difficult, however, also require high skill proficiency of providers for better patient outcomes.

The comparison of a Macintosh-blade shaped video laryngoscope and a hyperangulated video laryngoscope during 20 intubations, described the learning curve for the use of two different video laryngoscopes in children in a clinical setting [10]. This study concluded that the intubation times for attempts number 16–20 were significantly shorter for the Macintosh-blade shaped video laryngoscope than for the hyperangulated video laryngoscope. According to the authors expert paediatric anaesthesiologists rapidly acquire the skills needed for video laryngoscopes, even with limited experience, whereas our results show that anaesthesia residents do not handle this device that intuitively. Overall, it must be stated that with the relatively small number of intubations in this study, no trend can be depicted in terms of training effect for both KVL and DL, and thus no significant learning curve can be established.

Videolaryngoscopy has a distinct educational advantage over DL because both teacher and trainee have the same view at the same time. However, this benefit only occurs when the teacher provides a timely accurate, comprehensible, and targeted feedback.

The potential advantages of KVL compared to other laryngoscopes could be a shorter intubation time and better view of the glottis during intubation. Regarding intubation times, the results for TTBV in all comparable studies were a median of 5 s, and the view of the glottis was better when using the hyperangulated video laryngoscopes. The results of the present study confirm the improved glottic view for KVL with a C&L I rate of 80% (62% for DL). However, with a median of 7 s in both groups the time for TTBV was slightly longer than in the other studies. Furthermore, we observed that the time to glottic visualization is a less important parameter than the TTP because the quality of glottic visualization does not guarantee a rapid and successful intubation. Current literature indicates a median time to intubation for video laryngoscopy between 20 and 60 s [29]. These findings prove that intubation times are subject to great variability. One of the reasons might be the use of different definitions of intubation times, which is a common problem when comparing various studies on tracheal intubation [31]. Although the intubation times of the present study fit into the wide range of intubation times for video laryngoscopes mentioned above, we observed several specific factors that cause the prolongation of the intubation process in the KVL. First, the bulky design of the hyperangulated blade shape of the KVL complicates handling, especially for anaesthesia residents. Second, we used the recommended blade size for intubation of all children, regardless of weight and physiognomy. Considering that the manufacturer’s blade size recommendation for blade size 1 and size 2 overlaps for children aged 1 to 3 years, we suggest a more differentiated consideration in this age group. At this age, the pharyngeal space changes rapidly and therefore more attention should be paid to the blade size, as the introduction of a smaller blade might be easier given the hyperangulated design of the KVL.

The type of tube used could influence the results of studies comparing different intubation techniques. In the present study, the use of an armoured tube was almost 25% higher in the KVL group compared to the DL group. We assume that the placement of an armoured tube with a stylet carries the risk of the tube slipping out of the vocal cord area once the stylet is removed, resulting in a prolonged intubation time. However, to date there are no comprehensive data showing the impact of different tube types on the intubation process in paediatric airway management.

The results of this study should be interpreted in the context of the following limitations. First, only children with normal airways were enroled in this study, although the KVL with its hyperangulated shape was initially designed for the difficult airway. Therefore, these results cannot be extrapolated to children with difficult airways. Second, with the paucity of intubations performed in this study, a meaningful learning curve cannot be presented. Unfortunately, promising results on the performance of new airway devices obtained through manikin studies are not always validated by clinical trials. The clinical realization of this study was very complex, and the recruitment period took more than 24 months. Initially, it was planned that each resident was supposed to perform 10 intubations with each device. Even though the majority of trainees completed the 20 intubations, two were not able to complete the study as planned for the following reasons: administrative difficulties in obtaining written consent from both parents, children postponing surgery because of disease, residents rotating to other faculties. We therefore recruited one more than the anticipated 10 trainees in order to meet the study requirements, but when the corona pandemic started, recruitment came to a complete stop and the study was terminated and the available data was analysed.

In the era of supraglottic airway devices, infant intubation is not part of a resident’s daily routine anymore, so it is even more important to find a device with which one can learn the necessary skills as quickly and reliably as possible.

## 5. Conclusions

This study demonstrates that paediatric video laryngoscopy using the KVL performed by trainees leads to an increased rate of intubation failure with a prolonged intubation process. Furthermore, intubation using the KVL was considered difficult by more than half of the trainee anaesthesiologists. These results suggest that KVL should not be recommended as the primary method by which anaesthesia residents learn to intubate children.

## Figures and Tables

**Figure 1 jcm-11-05676-f001:**
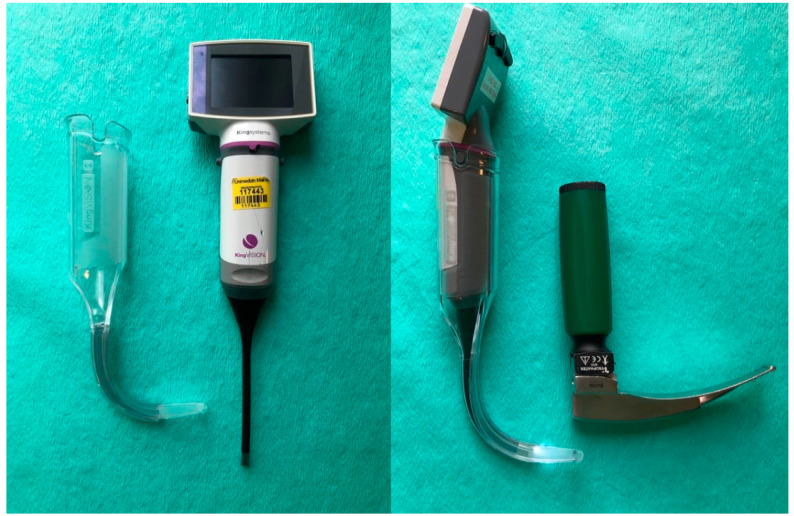
The King Vision™ Paediatric aBlade™ (Ambu^®^ GmbH, Bad Nauheim, Germany) video laryngoscope (KVL) blade size 2 (**left**) and the Macintosh laryngoscope (DL) blade size 2 (**right**).

**Figure 2 jcm-11-05676-f002:**
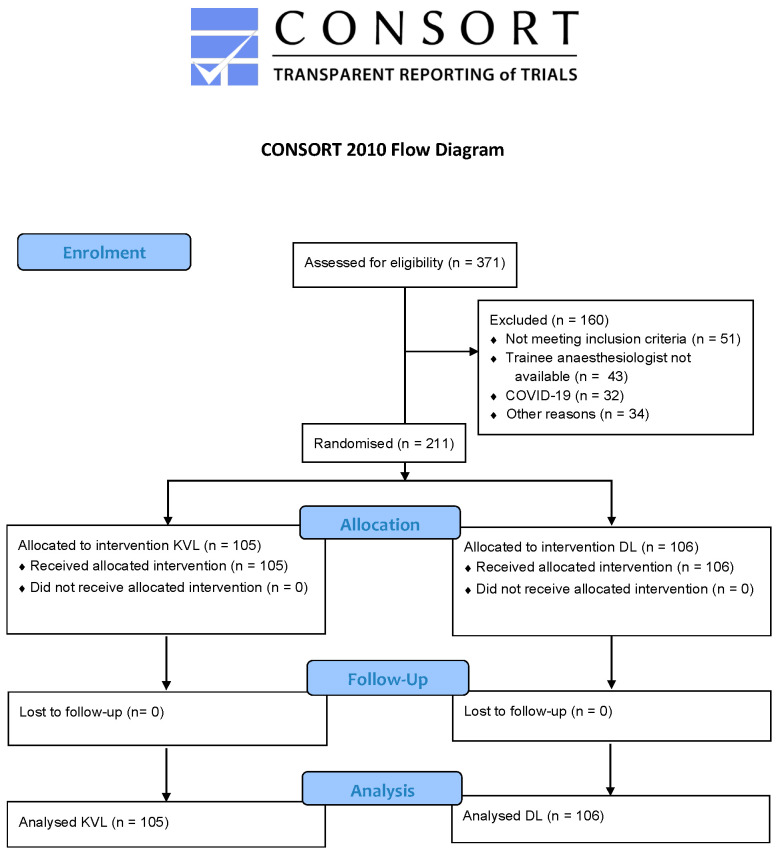
Consort flow diagram for patients in the study.

**Figure 3 jcm-11-05676-f003:**
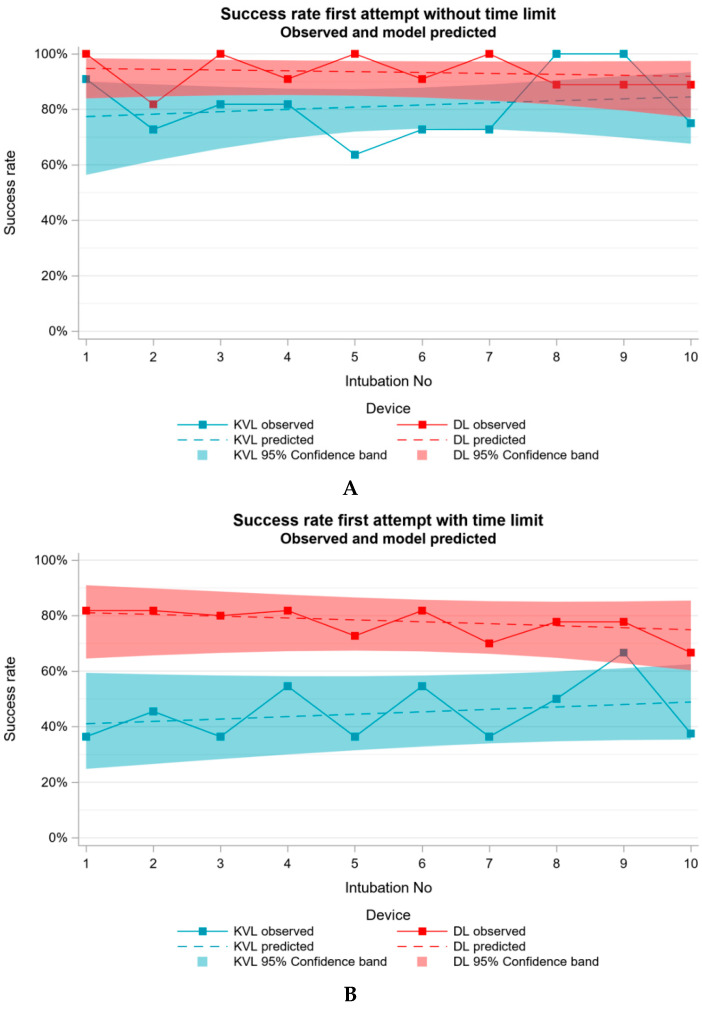
Observed learning curves and learning curves predicted from a generalized estimating equations logistic regression, which was performed to evaluate the progress of each trainee with increasing number of intubations. (**A**) Success rate without time limit, (**B**) with time limit.

**Figure 4 jcm-11-05676-f004:**
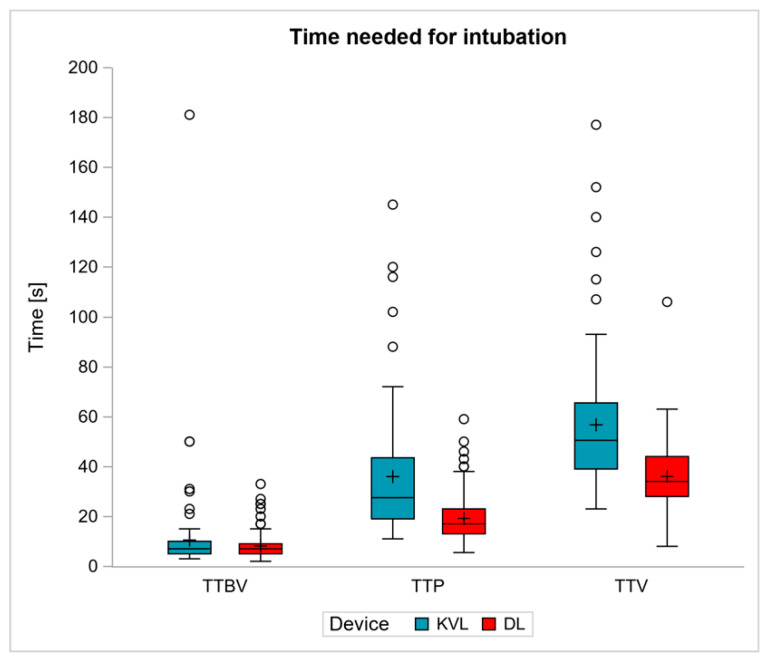
Time needed for the intubation steps in comparison of the King Vision™ Paediatric aBlade™ video laryngoscope (KVL) and direct laryngoscopy with the Macintosh blade (DL) in seconds [s]. TTBV: time to best view, TTP: time to placement of the tracheal tube, TTV: time to ventilation.

**Table 1 jcm-11-05676-t001:** Patient and anaesthesia resident characteristics in the King Vision™ Paediatric aBlade™ video laryngoscope (KVL) and conventional direct laryngoscopy (DL) group. Values are numbers (proportion) or median [IQR].

Patients	KVL	DL	*p*
	(n = 105)	(n = 106)	
Age; months	49 [35–69]	50 [35–73]	0.7402
Sex; male	67 (64%)	55 (52%)	0.0947
Weight; kg	17 [13–20]	17 [14–20]	0.7249
Height; cm	105 [95–115]	104 [95–117]	0.8196
ASA			0.1786
1	66 (63%)	77 (73%)	
2	37 (35%)	28 (26%)	
NA ^1^	2 (2%)	1 (1%)	
Type of surgery			0.0292
Otorhinolaryngology	90 (86%)	77 (73%)	
Others (pediatric surgery, ophthalmology)	15 (14%)	29 (27%)	
**Anaesthesia residents (n = 11)**			
Training status; months	49 [45–56]		
Practical year in anaesthesia	10/11 (91%)		
Clinical traineeship in anaesthesia	9/11 (82%)		
Age; years [range]	31 [30–33]		
Sex; male	4/11 (36%)		
Left-hander	1/11 (9%)		
Number of VL ^2^ in patients aged > 10 years			
<100	3/11 (27%)		
100–499	7/11 (64%)		
>500	1/11 (9%)		
Number of VL ^2^ in children aged < 10 years			
<10	6/11 (55%)		
10–49	5/11 (46%)		

^1^ NA = not applicable, ^2^ VL = video laryngoscopy. *p*-values derived from Wilcoxon test for age, weight, and height and Fisher’s test for sex, ASA classification, and type of surgery.

**Table 2 jcm-11-05676-t002:** Characteristics of the intubation attempt in the King Vision™ Paediatric aBlade™ video laryngoscope (KVL) and conventional direct laryngoscopy (DL) group. Values are numbers (proportion).

Intubation Attempt	KVL	DL	*p*
	(n = 105)	(n = 106)	
Oesophageal intubation at first attempt	2 (2%)	1 (1%)	0.9098
Timeout at first attempt	51 (49%)	19 (18%)	<0.0001
Tube type:			0.6262
Murphy tube	26 (25%)	42 (40%)	
Armoured tube	78 (74%)	63 (59%)	
Other	1 (1%)	1 (1%)	
Airway manoeuvres:			0.4079
OELM ^1^	10 (10%)	6 (6%)	
Position of head	6 (6%)	11 (10%)	
OELM ^1^ and position of head	4 (4%)	8 (8%)	
Cormack and Lehane Score I	84 (80%)	66 (62%)	<0.0001
Cormack and Lehane Score II	18 (17%)	38 (36%)	
Cormack and Lehane Score III	-	2 (2%)	
NA	3 (3%)	-	
POGO ^2^ Scale 100%	65 (62%)	46 (43%)	0.0495
Fogging of camera	57 (54%)	-	--
Complications (oesophageal intubation, desaturation > 2%, bleeding, no visualization)	39 (37%)	7 (7%)	0.2578
Difficulties (tube too big, tube placement not possible, guide wire problem, blade too big, visualization problem, lip injury)	36 (34%)	8 (8%)	<0.0001
Likert Scale rating usability of intubation:			<0.0001
Very easy (1)	13 (12%)	71 (67%)	
Easy (2)	32 (30%)	30 (28%)	
Difficult (3)	44 (42%)	4 (4%)	
Very difficult (4)	12 (11%)	-	
NA ^3^	4(4%)	1 (1%)	
Mean [95% CI] ^4^	2.5 [2.3;2.7]	1.4 [1.1;1.6]	<0.0001

^1^ OELM = optimal external laryngeal manipulation manoeuvre, ^2^ POGO = percentage of glottis opening, ^3^ NA = not applicable, ^4^ mean, CI and *p*-value derived from GEE model assuming normal distribution of Likert scale.

## Data Availability

Not applicable.

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
