# Peer review of "Video Laryngoscopy Using King Vision™ aBlade™ and Direct Laryngoscopy in Paediatric Airway Management: A Randomized Controlled Study about Device Learning by Anaesthesia Residents"

_jcm, 2022, doi:10.3390/jcm11195676_

Round 1
Reviewer 1 Report
The authors conducted an RCT in which they compared KVL and direct laryngoscopy in pediatric airway management. They showed that KVL was inferior to direct laryngoscopy in the first-attempt success rate. The manuscript is well written, and the topic is important since pediatric airway management is essential skill for anesthesia residents. However, I have several concerns for publication in the current form.
Major comment
According to the data on NCT03571295, predefined primary outcome of this study is “first attempt success rate of endotracheal intubation”. I guess time limit of intubation failure was not set in advance. The predefined primary oncome showed no statistical difference. Thus, the authors make it clear in the text and should tone down the conclusion.
Minor comments
1. I recommend to explain how sample size of this study was determined in the text.
2. I recommend to explain method for random allocation sequence in detail.
3. I recommend to apply same vertical scale in the Figure 3A and 3B.
4. According to the data on NCT03571295, 10 trainee anesthetists are scheduled to participate in the study. Actually, 11 trainee anesthetists are included. It is a minor change, though, I suppose readers want to explanation for this change.
Author Response
Dear Reviewer 1,
we are very grateful for your valuable commets and suggestions. We would like to adress all of your concerns. We highlighted all changes based on your suggestions in yellow within the manuscript (Changes concerning Reviewer 2 are highlighted in green). Please find our answers below:
Major comment
According to the data on NCT03571295, predefined primary outcome of this study is “first attempt success rate of endotracheal intubation”. I guess time limit of intubation failure was not set in advance. The predefined primary outcome showed no statistical difference. Thus, the authors make it clear in the text and should tone down the conclusion.
Thank you for this comment. We have double checked the NCT entry, and if you click on “tabular view” you can in fact see that we had preset a time limit there in 2018 when we started recruitment (also see ll. 110-112 in the manuscript). As outlined in ll. 114-116 of the manuscript, we had granted the supervising attendings the right to override the time limit if they felt that a potentially successful intubation would be aborted by it and in order to avoid unnessecary second intubation attempts. After completion of the recruitment process, we observed, that the preset time limit for KVL of 30 sec was in fact too short and couldn’t be sustained throughout the study without causing unnessecary second intubation attempts. We therefore changed the NCT entry to first attempt success rate in general, but did evaluate both outcomes.
For the predefined outcome, a statistically significant p-value was found based on a GEE model (line 178-180). For this reason, we feel that our conclusion simply highlights our findings.
Minor comments
- I recommend to explain how sample size of this study was determined in the text.
We have added an explanation how sample size of this study was determined in the materials and methods section (line 134-140):
“When determining sample size, we aimed to have at least 80% power to establish an overall 15% difference in success rates at the 5% significance level assuming that the lower success rate would be 80%. This would be possible with 76 independent observations per type of laryngoscope when using a chi-squared test. As multiple observations per provider would lead to dependent observations, thus losing some power, we decided to increase sample size to 100 observations per type of laryngoscope and a maximum of 10 observations per provider within each group.”
- I recommend to explain method for random allocation sequence in detail.
We have complemented your recommendation in our material and method section (line 88-91):
“The device for the first tracheal intubation of each trainee was randomly assigned to either the KVL or DL group using a web-based service (QuickCalcs, GraphPad Software, La Jolla, California, USA). Devices were then used in alternating order.”
- I recommend to apply same vertical scale in the Figure 3A and 3B.
We have replaced Figure 3A and 3B with a figure that shows a learning curve of the devives. That was recommeded by Reviewer 2.
- According to the data on NCT03571295, 10 trainee anesthetists are scheduled to participate in the study. Actually, 11 trainee anesthetists are included. It is a minor change, though, I suppose readers want an explanation for this change.
We explained this minor change in the limitation section (line 360-367):
“Initially, it was planned that each resident was should perform 10 intubations with each device. Even though the majority of trainees completed the 20 intubations, two were not able to complete the study as planned for the following reasons: administrative difficulties in obtaining written consent from both parents, children postponing surgery because of disease, residents rotating to other faculties. We therefore recruited one more than the anticipated 10 trainees in order to meet the study requirements but when the corona pandemic started, recruitment came to a complete stop and the study was terminated and the available data was analysed.”
Sincerely,
Dr. med. Nina Pirlich
Departement of Anaesthesiology
University Medical Center Mainz

Reviewer 2 Report
This is a well written paper regarding the use of advanced video laryngoscope (KVL) for intubation training in pediatric population.
This paper is of high educational interest since many speculations exist around newcoming devices not only ion anesthesia but in all medical specialties.
I have few remarks:
1. figures 1 and 2 - is not clear why it is expressed by individual residents? The result and discussion will benefit from adding a point on a learning curve. A learning curve KVL vs VL (sucess rate = f(device, n of attempts)) would better illustrate your findings
2. how many intubation was performed per resident?
3. please provide p values for comparisons in table 2
4. what does mean N/A for Likert scale in table 2 (L198)
5. You have more than 200 responses for Likers scale, you might treat statistics for Likert scale as for continuous data with an appropriate statistical test in addition to expressed response rates
Thank you
Author Response
Dear Reviewer 2,
we are very grateful for your valuable remarks. We would like to adress all of them. We highlighted all changes in green within the manuscript (Changes concerning Reviewer 1 are highlighted in yellow). Please find our answers below:
- Figures 1 and 2 - is not clear why it is expressed by individual residents? The result and discussion will benefit from adding a point on a learning curve. A learning curve KVL vs VL (sucess rate = f(device, n of attempts)) would better illustrate your findings
We followed your very valuable suggestion and instead of the previous figure, we decided to include the results from the generalized estimating equations logistic regression instead, which was performed to evaluate the progress of each trainee with increasing number of intubation to create a learning curve, combined with the observed proportion of success at each intubation number (new Figure 3 A und B).
We have included a corresponding explanation in the Materials and Methods section (line 143-145).
In addition, we have complemented the point “learning curve” in the discussion according to the obtained results (line 315-318).
- How many intubation was performed per resident?
We specified this information in the Materials and Methods section (line 86-88):
“Each trainee performed at least 7 intubations with the KVL and 6 direct laryngoscopies with the Macintosh blade (DL) but no more than a total of 20 intubations with both devices to assess his or her ability to successfully intubate a paediatric patient.”
Also check the yellow highlighted lines 360-367 (Answer Reviewer 1) in the Discussion for further explanation concerning the variing numbers.
“Initially, it was planned that each resident was supposed to perform 10 intubations with each device. Even though the majority of trainees completed the 20 intubations, two were not able to complete the study as planned for the following reasons: administrative difficulties in obtaining written consent from both parents, children postponing surgery because of disease, residents rotating to other faculties. We therefore recruited one more than the anticipated 10 trainees in order to meet the study requirements but when the corona pandemic started, recruitment came to a complete stop and the study was terminated and the available data was analysed.”
- Please provide p values for comparisons in table 2.
We have included p values for the comparison in table 1 and 2. p-values for age, weight and height are calculated via Wilcoxon test; p-values for sex, ASA classification and type of surgery are calculated via Fisher´s test. The p-values in table 2 are derived from GEE models with binary or multinominal endpoints and logit or cumulativ logit links. The models had device and intubation number as covariates and took into account that multiple observations per provider were included.
- What does mean N/A for Likert scale in table 2 (L198)
As outlined in the legend below table 2, N/A stands for not applicable, which means that data had not been documented by the performing resident.
- You have more than 200 responses for Likers scale, you might treat statistics for Likert scale as for continuous data with an appropriate statistical test in addition to expressed response rates.
Likert scale was now analysed via GEE model as normally distributed variables with identity link, device, and intubation number as covariables. We have once more adjusted the Materials and Methods section (line 152-154) with a corresponding explanation. P-values from this calculations have also been included in table 2 and its legend, including mean and CI.
Sincerely,
Dr. med. Nina Pirlich
Departement of Anaesthesiology
University Medical Center Mainz

Round 2
Reviewer 1 Report
Thank you for revising the manuscript. The authors answered all of my comments successfully, and the manuscript has improved.